# Band-Gap Solitons in Nonlinear Photonic Crystal Waveguides and Their Application for Functional All-Optical Logic Gating

**Vakhtang Jandieri [1], Ramaz Khomeriki [2,*], Tornike Onoprishvili [3], Daniel Erni [1], Levan Chotorlishvili [4], Douglas H. Werner [5] and Jamal Berakdar [4]**

[1] General and Theoretical Electrical Engineering (ATE), Faculty of Engineering, University of Duisburg-Essen and CENIDE— Center for Nanointegration Duisburg-Essen, D-47048 Duisburg, Germany; vakhtang.jandieri@uni-due.de (V.J.); daniel.erni@uni-due.de (D.E.)

[2] Physics Department, Tbilisi State University, 3 Chavchavadze, 0128 Tbilisi, Georgia

[3] School of Electrical and Computer Engineering, Free University of Tbilisi, 240 Agmashenebeli Ave., 0159 Tbilisi, Georgia; t.onoprishvili@freeuni.edu.ge

[4] Institut für Physik, Martin-Luther-Universität, Halle-Wittenberg, D-06099 Halle/Saale, Germany; levan.chotorlishvili@gmail.com (L.C.); jamal.berakdar@physik.uni-halle.de (J.B.)

[5] Department of Electrical Engineering, The Pennsylvania State University, University Park, PA 16802, USA; dhw@psu.edu

\* Correspondence: khomeriki@hotmail.com; Tel.: +995-595-397216

**Abstract:** This review paper summarizes our previous findings regarding propagation characteristics of band-gap temporal solitons in photonic crystal waveguides with Kerr-type nonlinearity and a realization of functional and easily scalable all-optical NOT, AND and NAND logic gates. The proposed structure consists of a planar air-hole type photonic crystal in crystalline silicon as the nonlinear background material. A main advantage of proposing the gap-soliton as a signal carrier is that, by operating in the true time-domain, the temporal soliton maintains a stable pulse envelope during each logical operation. Hence, multiple concatenated all-optical logic gates can be easily realized paving the way to multiple-input ultrafast full-optical digital signal processing. In the suggested setup, due to the gap-soliton features, there is no need to amplify the output signal after each operation which can be directly used as a new input signal for another logical operation. The efficiency of the proposed logic gates as well as their scalability is validated using our original rigorous theoretical formalism confirmed by full-wave computational electromagnetics.

**Keywords:** all optical logic gates; gap solitons; photonic crystal waveguides

## 1. Introduction

Several analytical and numerical methods have been proposed [1–3] to investigate the soliton propagation in various kinds of media [4]. In order to realize the propagation of solitons, the one- or quasi-one dimensionality of the medium is required. In the field of photonics, this can be achieved: by the refractive index varying between the guiding layer and the background medium [5], the photonic crystal (PhC) along (PhC fiber [6]) or perpendicular (PhC waveguide [7]) to the direction of the soliton propagation. In the manuscript, we are studying the planar PhC [8–14] waveguides (PCWs), which lead to a strong light confinement in the slow wave regime [15,16] and thus enhance the nonlinear effects [17–19]. If two or three PCWs are placed close to each other, coupled PCWs (C-PCWs) are formed [20].

In this review work, we provide an overview of our recent studies [21–25] about a realization of functional, compact and reliable all-optical logic gates, which may become key in modern (optical) computing and ultrafast optical signal processing [26,27]. We show a complete working scheme for a true all-optical NOT, an AND and a NAND logic gates. We are using an air-hole type hexagonal PhC [28,29] with a nonlinear silicon background material. In our studies, we are using PhCs [30–36], since they enable us to realize a dense

fully-optical signal processing on the chip level. A working concept is based on the virtually "perfect digitalization" of the time-domain signals inherent to the process of band-gap soliton transmission [21–23,37–41] in periodic nonlinear media [3,42]. In order to realize the 2 warnings "perfect digitalization", the operating frequency of the input signal should be chosen very close to the band edge of the PhC [21–25]. The temporal band-gap solitons with stable pulse envelopes are propagating in the C-PCWs, hence, no intermediate signal amplification between different C-PCW stages or even between different logic gates is needed. Since the logical operations can be realized using combinations of NAND logic gates, ultra-fast optical signal processing in future communication networks can be based on integrated all-optical logic gates with NAND functions [43].

This paper is organized as follows. In Section 2, we briefly present our original fast full-wave formalism of the modal analysis for periodic and band-gap structures [44–46] and apply it to the three C-PCWs in the linear regime. The derivation of the band-gap soliton solution is given in Section 3. An analytical formalism for the propagation of temporal solitons in PhC waveguides based on a multi-harmonic treatment of the nonlinear setup is presented. A rigorous expression for the nonlinear coefficient based on a modification of the refractive indices of each space-harmonic is derived. In Section 4, we investigate the ultra-compact functional all-optical logic gates using full-wave time-domain simulations of the pulsed digital signal processing. A full-wave computational electromagnetics analysis, namely the finite-difference time-domain (FDTD) method [47], is utilized to demonstrate the successful operation of functional all-optical logic gates. Conclusions are given in Section 5.

## 2. Formulation of the Problem

A well-established procedure for dealing with the nonlinearities is the variational Ansatz, from which a shape of the soliton is obtained [48,49]. Based on a multi-harmonic treatment using a full-wave modal analysis, we rigorously derive an expression for the nonlinear coefficient [21,24,25,37,38]. We perform numerical analysis for the gap-solitons propagation, because their formation time is relatively short and the registered gap soliton inside the guiding structure can be used for comparison with the analytical solution of the nonlinear Schrödinger equation.

As shown in Figure 1, we study three symmetric C-PCWs composed of a hexagonal lattice of circular air-holes periodically situated along the $x$-axis. The period of the structure is denoted by $h$ [24]. The guiding regions $(a)$, $(b)$ and $(c)$ with the same width $w$ are separated by the barrier layers. The number of the barrier layers is denoted by $N_B$. The radius of the air-holes is $r$, while $\epsilon_s = n_s^2$ is the relative dielectric permittivity of the background material. The electric field $E_x(x, y, t)$ in the nonlinear C-PCWs can be expressed as follows:

$$E_x = \frac{\Psi}{2} \sum_m (u_m^+ e^{ik_{ym}y} + u_m^- e^{-ik_{ym}y}) e^{i\beta_m x - i\omega t} + c.c. \tag{1}$$

where $k_{ym} = \sqrt{k_s^2 - \beta_m^2}$, $\Psi \equiv \Psi(x, t)$ is a slowly varying amplitude, $\beta_m = k_{x0} + \frac{2m\pi}{h}$, $k_s = \omega n_s \sqrt{\epsilon_0 \mu_0}$, $\omega$ is the angular frequency, $k_{x0}$ is the mode propagation constant along the $x$-axis, $k_{ym}$ is the transverse wavenumber of the $m$-th space-harmonic and "$c.c.$" represents the complex conjugate. We denote by $\mathbf{u}^+$ and $\mathbf{u}^-$ the vectors with components $\{u_m^+\}$ and $\{u_m^-\}$, respectively. Thus, $\mathbf{u}^+$ ($\mathbf{u}^+ = \mathbf{a}^+$, $\mathbf{b}^+$, $\mathbf{c}^+$) and $\mathbf{u}^-$ ($\mathbf{u}^- = \mathbf{a}^-$, $\mathbf{b}^-$, $\mathbf{c}^-$) are the amplitude vectors of the up-going and down-going space-harmonics along the $y$-axis in each guiding region. The expressions for the electric field components are written in the following form:

$$E_{xm}^{\pm} = \Psi u_m^{\pm}; \qquad E_{ym}^{\pm} = \mp \Psi \frac{\beta_m}{k_{ym}} u_m^{\pm} \tag{2}$$

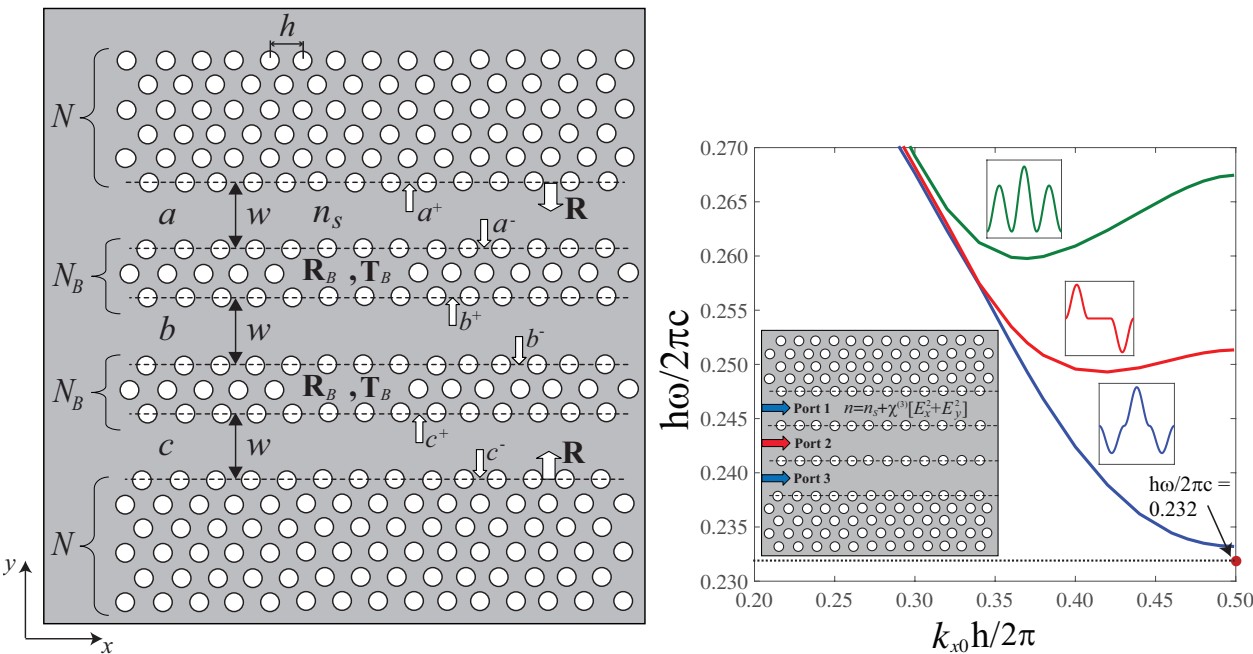

**Figure 1.** (**Left**) Three PhC waveguides with guiding regions (*a*), (*b*) and (*c*) having the same widths *w*. The waveguides are separated by the barrier layers with number of layers $N_B$. Planar PhCs are composed of a hexagonal lattice of the circular air holes periodically spaced along the *x*-axis with a period of *h*, which are infinitely long along the *z*-axis. $n_s$ is the refractive index of the background medium and radius of the air holes is *r*. Here, $a^+$ and $a^-$, $b^+$ and $b^-$, $c^+$ and $c^-$ define the up-going and down-going space-harmonics in the guiding regions (*a*), (*b*) and (*c*), respectively. (**Right**) Dispersion diagram of the even (blue and green curves) and the odd (red curve) modes for *H*-polarized field. The operating frequency is chosen as $\frac{h\omega}{2\pi c} = 0.232$ (red dot). Insets show the distributions of the magnetic field $H_z$ for the modes, as well as the geometry of the setup. Reprinted from Ref. [24].

The *x*-component of the electric displacement field $D_x$ is given by:

$$D_x = E_x \left[ \chi^{(1)} + \chi^{(3)} (E_x^2 + E_y^2) \right] \tag{3}$$

where $\chi^{(i)}$ is the *i*-th order optical susceptibility, and $\chi^{(1)} \equiv \epsilon_s = n_s^2$. Note that the second-order susceptibility term $\chi^{(2)} = 0$ [50]. Substituting the electric field components from Equations (1) and (2) into Equation (3) and grouping the terms containing the same $e^{i(\beta_m x - \omega t)} e^{\pm i k_{ym} y}$ *m*-th space harmonic, we obtain:

$$
\begin{aligned}
D_{xm}^{\pm} = \Psi u_m^{\pm} \Bigg[ & n_s^2 + \chi^{(3)} |\Psi|^2 \Bigg( \frac{3}{4} \left[ 1 + \frac{\beta_m^2}{k_{ym}^2} \right] |u_m^{\pm}|^2 + \frac{1}{2} \left[ 3 - \frac{\beta_m^2}{k_{ym}^2} \right] |u_m^{\mp}|^2 \\
& + \sum_{\mu \neq m} \frac{1}{2} \left[ 3 + \frac{\beta_\mu^2}{k_{y\mu}^2} \right] \left( |u_\mu^{\pm}|^2 + |u_\mu^{\mp}|^2 \right) + \sum_{\mu \neq m} \frac{\beta_m \beta_\mu}{k_{ym} k_{y\mu}} \left( |u_\mu^{\pm}|^2 - |u_\mu^{\mp}|^2 \right) \Bigg) \Bigg]
\end{aligned}
\tag{4}
$$

If the guided modes are well confined by the upper and lower PhCs with the number of layers *N* (*N* = 5), then $u_m^- \approx u_m^+$ and the last term in the square brackets of Equation (4) are very small. Thus, an expression for the refractive index can be given in the following form:

$$
\begin{aligned}
n_{m,\nu}^{\pm} = n_s + \frac{\chi^{(3)} |\Psi|^2}{4 n_s} \Bigg[ & \frac{3}{2} \left[ 1 + \frac{\beta_m^2}{k_{ym}^2} \right] |u_{m,\nu}^{\pm}|^2 + \left[ 3 - \frac{\beta_m^2}{k_{ym}^2} \right] |u_{m,\nu}^{\mp}|^2 \\
& + \sum_{\mu \neq m} \left[ 3 + \frac{\beta_\mu^2}{k_{y\mu}^2} \right] \left( |u_{\mu,\nu}^{\pm}|^2 + |u_{\mu,\nu}^{\mp}|^2 \right) \Bigg]
\end{aligned}
\tag{5}
$$

where $\nu = a, b, c$. From Equation (5) one can see that the refractive indices for the up-going and down-going $m$-th space harmonics are very close to each other $n_{m,\nu}^{+} \approx n_{m,\nu}^{-}$ and are proportional to the wave intensity $|\Psi|^2$, which leads to a shift in the angular frequency $\omega' = \omega + \delta\omega$ at a given mode propagation constant $k_{x0}$ due to the Kerr-type nonlinearity. The scattering amplitudes in each guiding region can be expressed through the reflection and transmission matrices characterizing the periodic structures in the following form:

$$\mathbf{a}^{-} = \mathbf{W}(\omega', k_{x0}, n_{m,a}^{\pm}) \mathbf{R}(\omega', k_{x0}, n_s) \cdot \mathbf{a}^{+} \tag{6}$$

$$\mathbf{a}^{+} = \mathbf{W}(\omega', k_{x0}, n_{m,a}^{\pm}) \left[ \mathbf{R_B}(\omega', k_{x0}, n_s) \cdot \mathbf{a}^{-} + \mathbf{T_B}(\omega', k_{x0}, n_s) \cdot \mathbf{b}^{+} \right] \tag{7}$$

$$\mathbf{b}^{+} = \mathbf{W}(\omega', k_{x0}, n_{m,b}^{\pm}) \left[ \mathbf{T_B}(\omega', k_{x0}, n_s) \cdot \mathbf{c}^{+} + \mathbf{R_B}(\omega', k_{x0}, n_s) \cdot \mathbf{b}^{-} \right] \tag{8}$$

$$\mathbf{b}^{-} = \mathbf{W}(\omega', k_{x0}, n_{m,b}^{\pm}) \left[ \mathbf{T_B}(\omega', k_{x0}, n_s) \cdot \mathbf{a}^{-} + \mathbf{R_B}(\omega', k_{x0}, n_s) \cdot \mathbf{b}^{+} \right] \tag{9}$$

$$\mathbf{c}^{+} = \mathbf{W}(\omega', k_{x0}, n_{m,c}^{\pm}) \mathbf{R}(\omega', k_{x0}, n_s) \cdot \mathbf{c}^{-} \tag{10}$$

$$\mathbf{c}^{-} = \mathbf{W}(\omega', k_{x0}, n_{m,c}^{\pm}) \left[ \mathbf{T_B}(\omega', k_{x0}, n_s) \cdot \mathbf{b}^{-} + \mathbf{R_B}(\omega', k_{x0}, n_s) \cdot \mathbf{c}^{+} \right] \tag{11}$$

with

$$\mathbf{W}(\omega', k_{x0}, n_{m,\nu}^{\pm}) = [e^{ik_{ym}w}], \qquad \nu = a, b, c \tag{12}$$

where $\mathbf{W}(\omega', k_{x0}, n_{m,\nu}^{\pm})$ defines the phase shift of each $m$-th up-going and down-going space harmonic within the guiding regions and it is a diagonal matrix, $\mathbf{R}(\omega', k_{x0}, n_s)$ is the generalized reflection matrix for the upper and lower $N$-layered PhCs, while the generalized reflection and transmission matrices of the PhC-s barriers are $\mathbf{R_B}(\omega', k_{x0}, n_s)$ and $\mathbf{T_B}(\omega', k_{x0}, n_s)$, respectively. We use our originally developed formalism [44,45] to rigorously calculate the generalized reflection and transmission matrices of the PhCs. The fast and self-contained formalism enables us to rigorously analyze the electromagnetic scattering, guidance and radiation in periodic and band-gap structures in a short computation time [44,45].

The linear system of Equations (6)–(11) can be rewritten as follows:

$$\mathbf{\Omega}(\omega', k_{x0}, n_{m,\nu}^{\pm}) \cdot \mathbf{A}^{T}(\omega', k_{x0}, n_{m,\nu}^{\pm}) = 0 \tag{13}$$

where $\mathbf{A} = \left\{ [...a_{-m}^{+}, ..., a_m^{+}...], ..., [...b_{-m}^{-}, ..., b_m^{-}...], .. \right\}$, "$T$" represents the transpose of the vector, $\mathbf{\Omega}$ is a block matrix and the size of each sub-matrix is $(2M+1) \times (2M+1)$. Expanding Equation (13) in terms of the perturbation $\delta\omega$ and $|\Psi|^2$, and vector multiplying with $\mathbf{A}'$ from the left while considering that $\mathbf{A}' \cdot [\mathbf{\Omega}(\omega, k_{x0}, n_s)] = 0$, the following expression can be obtained:

$$\delta\omega \mathbf{A}' \frac{\partial \mathbf{\Omega}(\omega, k_{x0}, n_s)}{\partial \omega} \mathbf{A}^{T} = -|\Psi|^2 \mathbf{A}' \frac{\partial \mathbf{\Omega}(\omega, k_{x0}, n_{m,\nu}^{\pm})}{\partial(|\Psi|^2)} \mathbf{A}^{T} \bigg|_{n_{m,\nu}^{\pm}=n_s}. \tag{14}$$

The eigenvectors $\mathbf{A}'$ and $\mathbf{A}$ are numerically calculated for the truncated block matrix $\mathbf{\Omega}(\omega, k_{x0}, n_s)$ after confirming the convergence of the solutions that satisfy the condition $\det[\mathbf{\Omega}(\omega, k_{x0}, n_s)] = 0$. The second term in Equation (14) can be rewritten as:

$$\frac{\partial \mathbf{\Omega}(\omega, k_{x0}, n_{m,\nu}^{\pm})}{\partial(|\Psi|^2)} = \sum_{m,\nu} \frac{\partial(n_{m,\nu}^{\pm})}{\partial(|\Psi|^2)} \frac{\partial \mathbf{\Omega}(\omega, k_{x0}, n_{m,\nu}^{\pm})}{\partial(n_{m,\nu}^{\pm})} \bigg|_{n_{m,\nu}^{\pm}=n_s}$$

Note that $\frac{\partial(n_{m,\nu}^{\pm})}{\partial(|\Psi|^2)}$ can be derived directly from Equation (5), while $\frac{\partial \mathbf{\Omega}(\omega, k_{x0}, n_{m,\nu}^{\pm})}{\partial(n_{m,\nu}^{\pm})} \bigg|_{n_{m,\nu}^{\pm}=n_s}$ should be numerically calculated for each $m$-th space-harmonic. The calculation method follows the approach presented in [44,45], albeit a slight modification of Equations (6)–(11) is needed by implementing the Fresnel matrices which are diagonal (Figure 2).

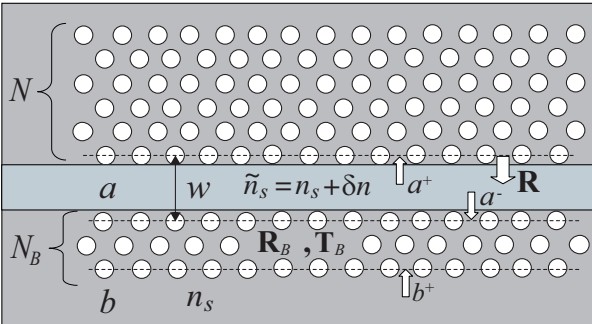

**Figure 2.** A single PhC waveguide with a guiding region ($a$) having a refractive index $\tilde{n}_s = n_s + \delta n$, while the refractive index of the other regions is the same as in Figure 1, i.e., it is equal to $n_s$. Reprinted from Ref. [24].

## 3. Soliton Solution for Coupled Nonlinear Photonic Crystal Waveguides

Using the well-established method presented in [51–53], the following expression for the slowly varying amplitude $\Psi(x, t)$ is obtained:

$$i\left(\frac{\partial \Psi}{\partial t} + v_g \frac{\partial \Psi}{\partial x}\right) + \frac{\omega''}{2}\frac{\partial^2 \Psi}{\partial x^2} + \gamma |\Psi|^2 \Psi = 0 \tag{15}$$

where $\gamma = -\frac{\partial(\delta\omega)}{\partial(|\Psi|^2)}$ is a nonlinear coefficient derived from Equation (14), the group velocity $v_g = \frac{\partial\omega}{\partial k_{x0}}$ and the group velocity dispersion $\omega'' = \frac{\partial^2\omega}{\partial k_{x0}^2}$ can be obtained from the dispersion curve. Then, the solution for the slowly varying amplitude $\Psi(x, t)$ can be written in the following form:

$$\Psi(x, t) = \frac{Fe^{-i(\Delta\omega)t}}{\cosh\left[(x - v_g t)/\Lambda\right]} \tag{16}$$

where

$$\Lambda = \sqrt{\frac{\omega''}{\gamma F^2}}, \qquad \Delta\omega = -\frac{\gamma F^2}{2} = \frac{F^2}{2}\frac{\partial(\delta\omega)}{\partial(|\Psi|^2)} \tag{17}$$

Note that $\Lambda$ represents the width of the temporal soliton, $\Delta\omega$ is the shift of the angular frequency due to Kerr nonlinearities and $F$ denotes the amplitude of the soliton. Finally, substituting Equation (16) into Equations (1) and (2), we obtain the expressions for the electric fields from which the magnetic field can be calculated.

As a numerical experiment, we consider three air-hole type C-PCWs located in a dielectric background medium with a linear refractive index $n_s = 2.95$ (crystalline silicon) in conjunction with a Kerr-type nonlinearity [22]. The thicknesses of the upper and lower PhCs is taken as $N = 5$ and the radius of the air-holes is $r = 0.32h$ ($h$ is the period of the PhCs). The number of the barrier layers is $N_B = 1$, and the length of the PhC is $30h$. The dispersion curve is illustrated in Figure 1 and we are operating at the normalized frequency $\frac{h\omega}{2\pi c} = 0.232$ (Figure 1). A continuous wave (CW) signal with ($H_z, E_x, E_y$) is injected through the middle waveguide (guiding region ($b$)). The injected peak power of the CW signal is $\chi^{(3)}E_0^2 = 0.1410$ and $n_2 = 3 \cdot 10^{-18}[\text{m}^2 \cdot \text{W}^{-1}]$, $E_0^2 = 2.7 \cdot 10^{18}[V^2 \cdot \text{m}^{-2}]$. Figure 3a demonstrates the magnetic field distribution of the gap soliton propagation. From the numerical simulations, it follows that the amplitude $F$ and the width $\Lambda$ of the gap soliton are around $F = 300$ A/m and $\Lambda = 4.10h$, respectively. Figure 3b shows the dependence of the magnetic field $H_z$ versus the dimensionless parameter $x/h$ at $y = 0$ (see Figure 3a), where [a.u] denotes arbitrary units. Numerical results from the FDTD analysis are indicated by the blue line, while the theoretical result is shown by a dashed red line. The width of the band-gap soliton obtained from Equation (17), is found to be $\Lambda = 3.95h$, which is in a very good agreement with the result obtained based on FDTD

($\Lambda = 4.10h$). The scattering amplitudes are calculated by solving the eigenvalue problem defined as $\mathbf{\Omega}(\omega, k_{x0}, n_s) \cdot \mathbf{A}^T = 0$. Figure 3c shows the dependence of the magnetic field $H_z$ versus $y/h$ at $x = 0$ (see Figure 3a). The result shown by a red dotted line is obtained using our original method [44,45], whereas a blue line demonstrates the result based on FDTD. An excellent agreement is observed between these results.

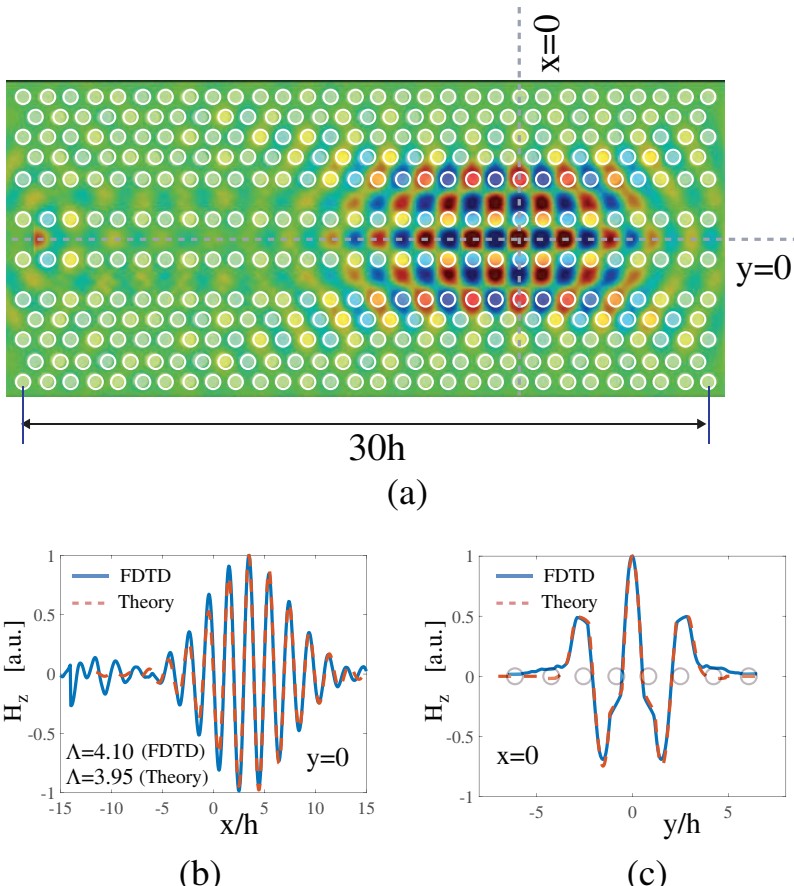

(a)

(b)                                      (c)

**Figure 3.** (**a**) The magnetic field of the gap soliton when a continuous signal with the peak power $\chi^{(3)} E_0^2 = 0.1410$ is injected into the guiding region (*b*). The operating frequency is $\frac{\hbar \omega}{2\pi c} = 0.232$ (Figure 1). (**b**) Magnetic field distribution $H_z$ versus the non-dimensional parameter $x/h$ at $y = 0$ (Figure 3a), where the blue line shows the numerical result based on FDTD analysis, while the red dashed line represents the result using Equation (1) together with Equations (6)–(11), (16) and (17). (**c**) Magnetic field $H_z$ versus $y/h$ at $x = 0$ (Figure 3a), where the blue line represents the numerical result based on FDTD analysis and the red dashed line shows the theoretical analysis based on Equation (1) together with Equations (6)–(11). Here [a.u] stands for arbitrary units. Reprinted from Ref. [24].

## 4. Numerical Simulations on Functional All-optical Logic Gates and Discussions

We examine a dielectric background material with a linear refractive index $n_s = 2.95$ and the medium is characterized by a Kerr-type nonlinearity. A 2D hexagonal lattice of air holes (the radius of the air-holes is $r = 0.32h$, $h$ is the period of the lattice) is implemented, thus forming the planar PhC. In order to achieve a strong confinement of the modes, it is sufficient to have $N = 4$ rows along the $y$-axis and the propagation constant $k_{x0}$ is a pure real value [44]. For practical application, we suggest to use the materials with enhanced third-order nonlinearities such as hybrid organic- *Si*-on-insulator compounds [50]. It should be mentioned that the original planar 3D structure [28,29] can be very well approximated by our 2D model. Its band-gap for the $H$-polarized field is located within a frequency range of $0.230 < \frac{\hbar \omega}{2\pi c} < 0.310$, where $c$ is the speed of light. We have calculated

the dispersion curves (Figure 1b) for three different modes of the three C-PCWs (blue and green lines correspond to symmetric modes and the red line describes an antisymmetric mode) based on our original method briefly described in Section 2. The operating frequency $\frac{h\omega}{2\pi c} = 0.232$ is indicated by the red dot. We undertake a comprehensive full-wave computational electromagnetics analysis based on the FDTD method and use perfectly matched layer (PML) boundaries for truncating the simulation domain [47]. FDTD simulations correspond to spatio-temporal discretization with $\Delta x = \Delta y = 8.611 \cdot 10^{-8}$ (m) and $\Delta t = 1.9295 \cdot 10^{-16}$ (s), respectively. In the numerical investigations, we do not consider attenuation effects, i.e., PhC propagation loss. According to the experimental data in [18], propagation loss inside a single PCW is only around 8–9%.

In our proposed working scheme for all-optical logic gates, the operating frequency $\frac{h\omega}{2\pi c} = 0.232$ is located at the edge of the dispersion curve of the symmetric mode (blue line in Figure 1). It is obvious that in the linear regime none of the modes are excited. In the weakly nonlinear regime, we observe a small frequency shift, and thus, we are only interested in the symmetric mode (blue line in Figure 1). The dispersion diagram of that symmetric mode can be well approximated by a parabola (i.e., the second derivative of the angular frequency with respect to the propagation constant is positive) [19,24,54]. In all suggested logic NOT, AND and NAND devices, we exploit three planar nonlinear C-PCW-s. Note that for the NOT logic gate we have one gate Port (Port 1), while for the AND and NAND logic gates we use two gate Ports, namely Port 1 and Port 3. Input Port 2 is used for CW signal, and we always monitor the output signal at the end of Port 2.

### 4.1. All-Optical NOT Logic Gate

The realization of an all-optical NOT logic gate is demonstrated in Figures 4 and 5. We inject CW (see Figures 4b and 5b), with an amplitude of $A = 0.956$, which is well below the band-gap transmission threshold $A = 1.1$, into the middle Port 2. We also launch a train of five Gaussian pulses with pulse repetition time of 15.43 ps, an amplitude of $A = 0.812$ and the full duration at half maximum (FDHM) of 2.22 ps into Port 3 as illustrated in Figure 4c. If the carrier oscillations of the Gaussian pulses are in phase with that of the CW signal, the signal amplitude overcomes the threshold and a train of temporal band-gap solitons will be formed. This scenario holds if the carrier oscillations of the Gaussian pulses are in phase with the CW signal within a range of approximately $\pm 20°$. The phase mismatch will affect the band-gap soliton formation time, and therefore, the phase difference should be minimized.

Figure 4d,e describe the magnetic field profile of one of the propagating temporal solitons at different moments in time and Figure 4f shows the output signals—a train of five temporal solitons—at the end of Port 2. Their amplitudes and widths are $A = 0.52$ and 2.22 ps, respectively. The registered output signal characteristics do not change depending on the input Gaussian pulse amplitudes as displayed in Figure 4g. This effect we call perfect digitalization. This mechanism is useful for all optical digital signal processing operations when an output signal is used as a new input signal for another logic gate operation. Next, in order to explicitly demonstrate the functionality of the all-optical NOT logic gate (i.e., output "0" at input "1"), we inject three Gaussian pulses into Port 1 (see Figure 5a) with the FDHM pulse width and pulse repetition time equal to 2.22 ps and $2 \cdot 15.43 = 30.86$ ps, respectively, and the amplitudes are $A = 0.52$. All other input characteristics are exactly the same as in the previous case for Ports 2 and 3. At the same time, the carrier oscillations of the Gaussian pulses through Port 1 and Port 3 are in counter phase and as a result, only two temporal band-gap solitons are observed at the end of the C-PCWs (Figure 5e). The other three input Gaussian pulses into Port 3 are compensated out by counter phase pulses into Port 1 due to destructive interference between them and no gap solitons are formed.

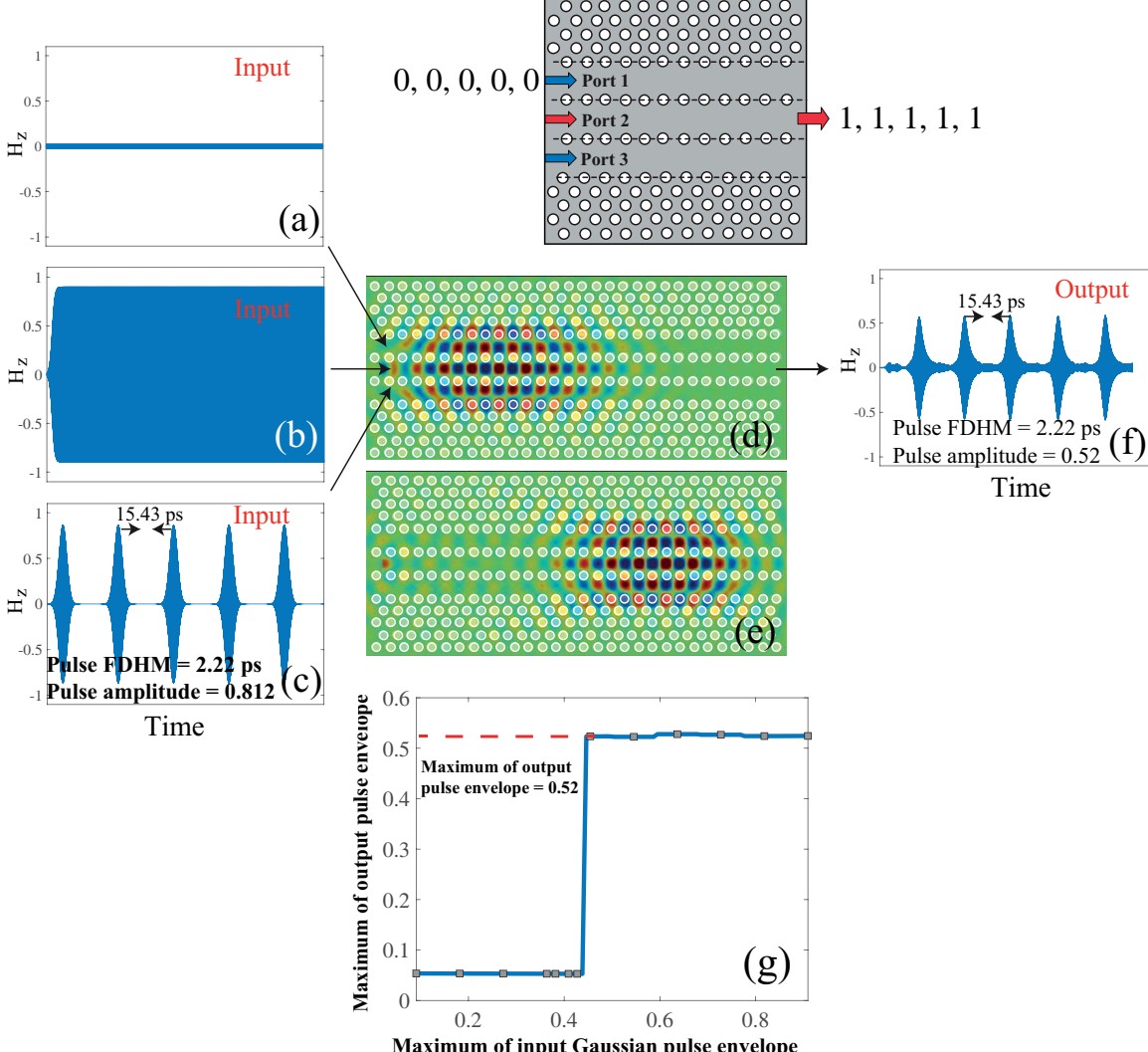

**Figure 4.** Results of simulations of all-optical NOT logic gate: in graphs (**a**–**c**), we demonstrate the injected signals: no signal into Port 1, a CW signal with amplitude $A = 0.956$ into Port 2 and a train of five Gaussian pulses with amplitudes $A = 0.812$, FDHM of 2.22 ps and a pulse repetition time of 15.43 ps into Port 3, respectively. In graphs (**d**,**e**), the magnetic field $H_z$ distributions of the band-gap soliton are illustrated and in graph (**f**) the magnetic field $H_z$ of the received signal at a distance $x = 30h$ is plotted. In graph (**g**), the dependence of the maximum of output pulse envelope versus the maximum of the input Gaussian pulse envelope is given. The peak level contrast between "1" and "0" is 20 dB. Reprinted from Ref. [25].

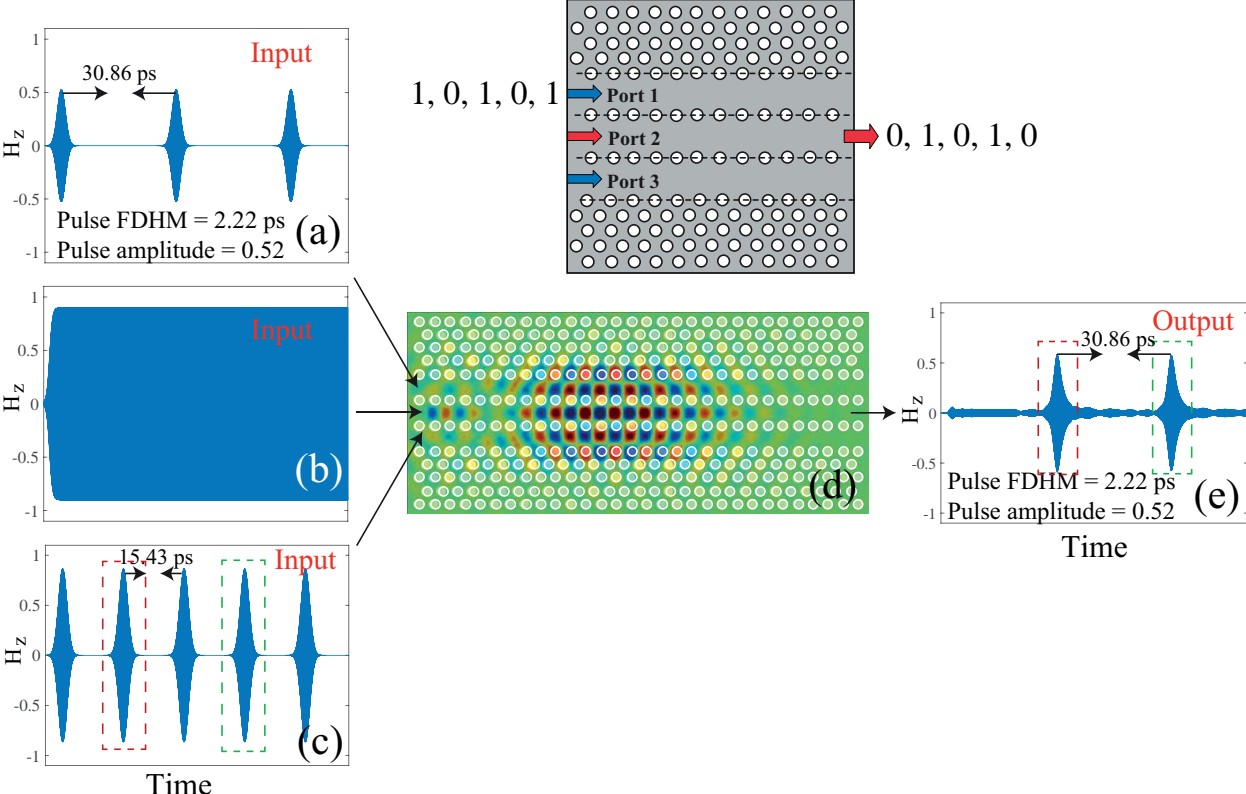

**Figure 5.** All-optical NOT logic gate: (**a**–**c**) display the pulses injected into the Ports: three Gaussian pulses with pulse repetition time of $(2 \cdot 15.43) = 30.86$ ps, FDHM of 2.22 ps and amplitude $A = 0.52$ are injected into Port 1; a CW signal with amplitude $A = 0.956$ is injected into Port 2; a train of five Gaussian pulses with pulse repetition time of 15.43 ps, the FDHM of 2.22 ps and amplitude $A = 0.812$ is injected into Port 3. Graph (**d**) shows the magnetic field profile of the propagating signal pulse and graph (**e**) displays the magnetic field $H_z$ of the output signal.

### 4.2. All-Optical AND Logic Gate

A setup of a functional all-optical AND logic gate is represented in Figure 6. A CW signal is injected into the Port 2 with the same amplitude $A = 0.956$ as in case of the NOT logic gate. The upper and lower sets of graphs in Figure 6 represent different scenarios of injection of signals into Port 1. Particularly, in graph (a) of the upper set, no signal is launched into Port 1, while in graph (a) of the lower set, a train of five Gaussian pulses is injected into the same Port 1. In graphs (c), a train of five Gaussian pulses with amplitude $A = 0.52$, FDHM of 2.22 ps and pulse repetition time of 15.43 ps is injected into Port 3. In graphs (d) and (e), magnetic field distributions of the signal pulses, i.e., the band-gap solitons, propagating in the C-PCWs at different moments in time are shown. In graphs (f), magnetic fields $H_z$ of the output signal at a distance $x = 30h$ associated with Port 2 are depicted. We demonstrate the operation of the all-optical AND gate when by "1" and "0" as the input pulses we get "0" as the output signal (upper set of graphs). In the lower set of graphs, the operation of the all-optical AND gate is shown when by "1" and "1" as input pulses we obtain "1" as the output signal. In the latter case, the pulses in Port 1 and Port 3 are in phase with the carrier oscillations of the CW signal.

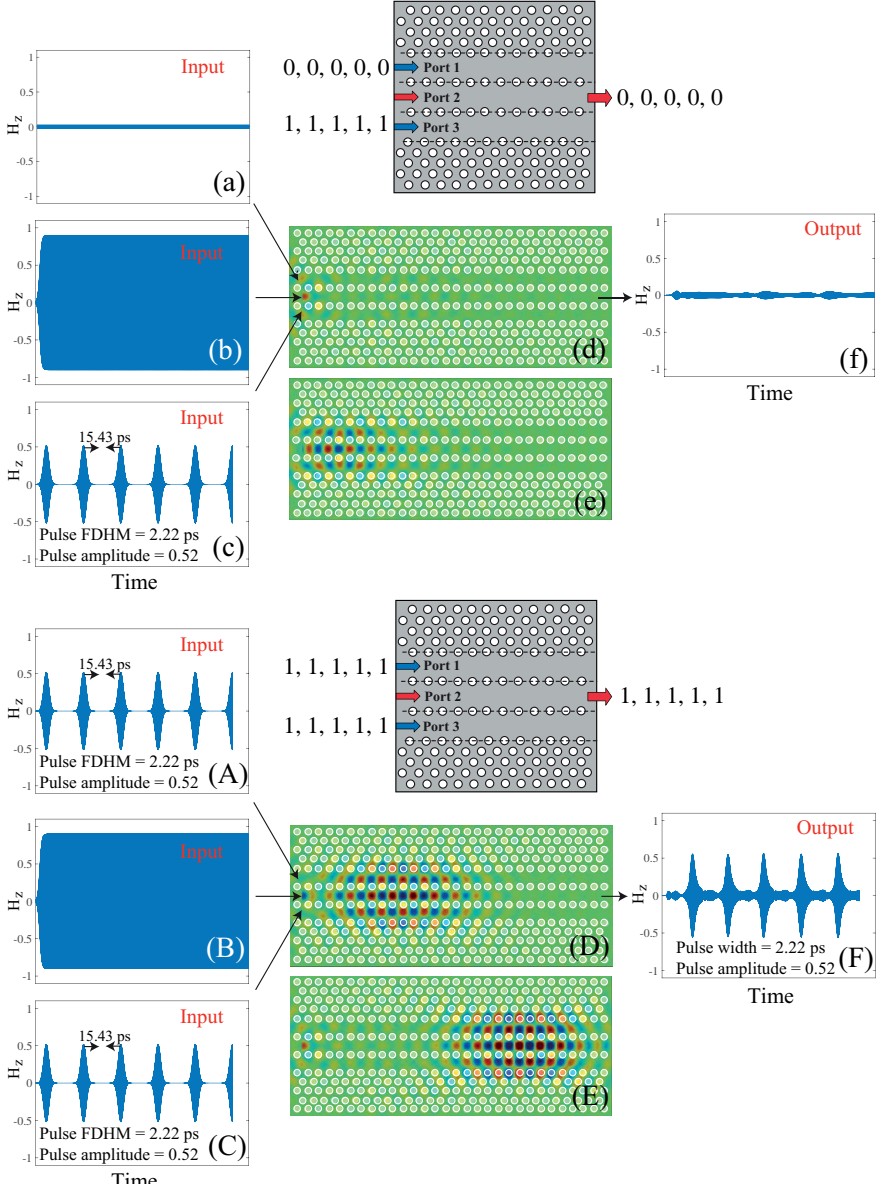

**Figure 6.** Realization of a functional all-optical AND logic gate: the upper and lower set of graphs represent different scenarios of injection of the input signals into Port 1. Upper graph: (**a**) no signal is injected into Port 1; (**b**) a CW signal is injected into Port 2; (**c**) a train of five Gaussian pulses is launched into Port 3; (**d**) and (**e**) magnetic field distributions inside the C-PCWs at different moments; (**f**) magnetic field $H_z$ of the received signal at a distance $x = 30h$ associated with Port 2. Lower graph: (**A**) a train of five Gaussian pulses is launched into Port 3; (**B**) a CW signal is injected into Port 2; (**C**) a train of five Gaussian pulses is launched into Port 3; (**D**) and (**E**) magnetic field distributions of the signal pulses; (**F**) magnetic field $H_z$ of the received signal. The peak level contrast between "1" and "0" amounts to 20 dB. Reprinted from Ref. [25].

### 4.3. All-Optical NAND Logic Gate

For the realization of a NAND gate, we connect in series an AND and a NOT gate. Figures 7 and 8 represent the corresponding operations "1", "0" and "1", "1" of the NAND logic gate on a single (enlarged) PhC chip. The output signal from the Port 2 of the AND gate is used as input signal in Port 1 of the NOT gate. This is possible due to the "perfect digitalization" of the temporal band-gap soliton signal created due to band-gap transmission (see Figure 4g). Based on the extensive computational simulations, the output signals are retrieved (see Figures 7d and 8c) showing the output of the respective operation

of the NAND all-optical gate. A challenging point of our NAND gate realization on a single chip is the "bridge" section. In the "bridge" section, we have decreased the radii of the air-holes (depicted by red in Figures 7 and 8) and removed a certain number of the air-holes in order to avoid a substantial decrease of the output signal amplitude from the AND gate. Particularly, the radii of the air-holes in the "bridge" section are taken as 0.18*h*. By decreasing the radii of the air-holes, the dispersion diagram given in Figure 1b is shifted towards the lower frequencies (the relevant analysis has been undertaken in [44]). Therefore, the output pulse from the Port 2, which looses a portion of power due to back-reflections, can pass through the NOT gate. We have considered a number of possibilities to decrease back-reflections, namely by properly adjusting the radii and position of the air-holes [55] in the "bridge" section. This is crucially important since the back-reflections can lead to creation of unwanted temporal gap-solitons in the AND gate. Hence, further detailed optimization is needed for our NAND single chip device. Another possibility to further decrease the back-reflections in the "bridge" section is to insert a so called "wave dump" that guides the reflected pulse from the NOT gate out of the structure into free space [55].

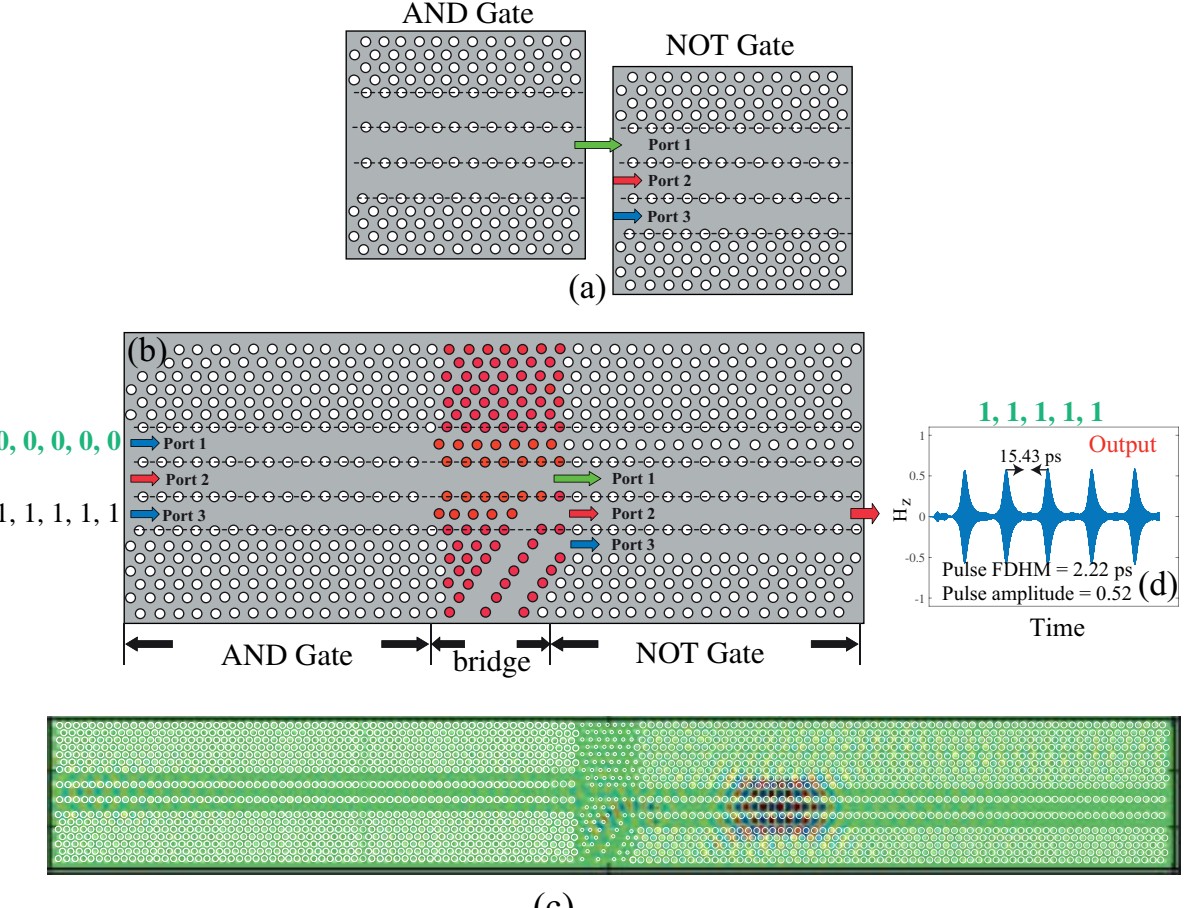

**Figure 7.** (**a**) Schematic visualization of NAND logic gate as a series connection of the AND and the NOT logic gate; (**b**) NAND logic gate on a single enlarged chip; (**c**) magnetic field profile of the gap soliton pulse in the NAND logic gate; (**d**) output magnetic field $H_z$ from the Port 2 after injection "0" and "1" input signals into the Ports 1 and 3, respectively. Reprinted from Ref. [25].

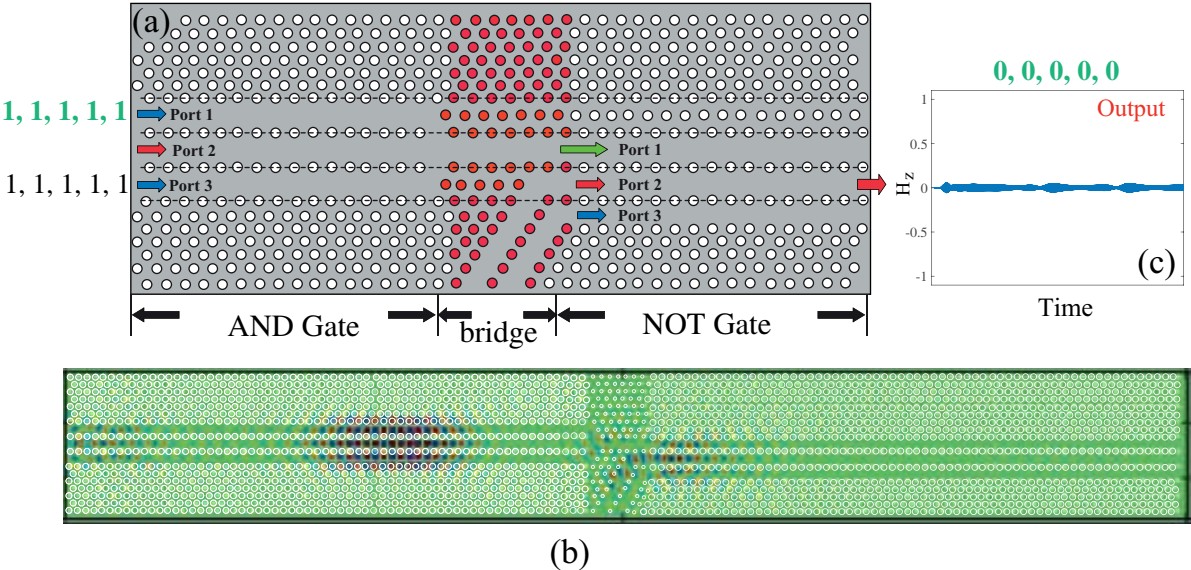

(b)

**Figure 8.** The same as in Figure 7, but the output magnetic field $H_z$ is displayed after injection of "1" and "1" input signals into the Ports 1 and 3, respectively. (**a**) Output signal from the AND logic gate is injected into Port 1 of the NOT logic gate; (**b**) magnetic field distribution of the signal pulses in the NAND logic gate; (**c**) magnetic field $H_z$ of the received signal. Reprinted from Ref. [25].

## 5. Conclusions

We have suggested a functional all-optical scheme for compact NOT and AND logic gates in a realistic model of Kerr-type nonlinear C-PCWs by using the dynamical properties of optical band-gap solitons. Moreover, we have built a NAND all-optical logic gate as a series connection of AND and NOT logic gates on a single chip using a special design of the "bridge" section between AND and NOT logic gates. The key element of our proposed concept behind the all-optical operations is the characteristic "perfect digitalization" of the emitted band-gap solitons. We believe that these studies could serve as a practical methodology for designing ultra-compact all-optical devices working with nonlinear optical signals, which might be applicable, e.g., for high-performance parity-bit checking [56]. We have presented a complete theoretical and numerical study, but we have not conducted experimental studies. The authors hope that this paper might motivate the experimental groups and that they succeed in the realization of the all-optical logic gates based on the proposed "perfect digitalization" effect.

**Funding:** This work was supported by Shota Rustaveli National Science Foundation of Georgia (SRNSFG) (Grant No. FR-19-4058). This work was partly supported by the Deutsche Forschungsgemeinschaft (DFG, German Research Foundation)—CRC/TRR 196 MARIE under Grant 287022738 (project M03).

**Conflicts of Interest:** The authors declare no conflicts of interest.

## Abbreviations

The following abbreviations are used in this manuscript:

| | |
|---|---|
| PhC | Photonic crystal |
| PCW | Photonic crystal waveguide |
| C-PCW | Coupled photonic crystal waveguide |
| CW | Continuous wave |

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
