# Peer review of "Band-Gap Solitons in Nonlinear Photonic Crystal Waveguides and Their Application for Functional All-Optical Logic Gating"

_photonics, doi:10.3390/photonics8070250_

Round 1

Reviewer 1 Report

This review paper summarizes the authors’works on band-gap temporal optical solitons propagating in the photonic crystal slab where waveguides are created and Kerr-type focusing nonlinearity is assumed. NOT, AND and NAND logic operations are demonstrated with the solitons. These demonstrations are of potential applications in the future light-based information transportation and processing. I certainly support the publication of this review in Photonics.

An only weakness of the paper is that it does not include experimental results on this topic. I suggest include, at least, some discussions on the possible experimental demonstrations of these logic operations, especially, in terms of the magnitude of the nonlinear effect that is required.

Author Response

It is quite difficult to conduct the experiments for nonlinear processes, since we need high power sources.
However, we hope that our idea and complete theoretical and numerical studies for realization of all-optical
logic gates might motivate the experimental groups. There are some experimental works on soliton
propagation in the nonlinear Kerr-type medium, but the most of them are focused on the single waveguide
(not a realization of an all-optical logic-gate). We have added a few sentences in the Conclusion.

Reviewer 2 Report

This is a very well written paper on band-gap solitons in nonlinear photonic crystal waveguides and relevant applications in all-optical logic gating.

The results shown in the paper are based on a rigorous full-wave approach presented in [44,45] in conjunction with a  well-established procedure for dealing with nonlinearities, as shown in Sections II and III. Results in Fig. 3 validate the approach. Furthermore, the concept of the virtually "perfect digitalization" of the time-domain signals is used to design functional all-optical logic gates.

The reviewer believes that a brief comment on the limits, if any, of the presented approach, compared with those proposed in the literature, should be addressed in the paper. Also future possible developments and perspectives of the described method and analysis can be provided in this review paper.

Author Response

The final step of this work is experiment, of course. We have presented a complete theoretical and numerical
study, but we have not conducted experimental studies. But the authors hope that this paper might motivate the experimental groups and they succeed in the realization of the all-optical logic gates based on the proposed "perfect digitalization" effect. We have added a few sentences in the Conclusion.

Reviewer 3 Report

The authors review band-gap solitons in nonlinear photonic crystal waveguides and discuss their application for functional all-optical logic gating. The paper is well-written and clear.
The following points must be clearly addressed before publication.

The literature review in Introduction can be improved. What about other linear and nonlinear techniques? advantages and disadvantages?
It is said: (In order to realize the propagation of solitons, the one- or quasi-one dimensionality of the medium is required). Elaborate on this.
What about other nonlinearities in silicon? Nonlinear loss, two photon absorption?
It is said: (Note that the second-order susceptibility term χ(2) = 0 [50]). Why? Is it really zero or small? Note you have a structured silicon.
Is the equation correct in the inset of Fig.1 (Right)?
In Eq. 15 what about the loss, higher-order dispersion terms and other 3rd-order nonlinearities (Raman and so on)?
Fig. 3b: why is Lambda different for two cases? Is it a very good agreement (line 142)?
Fig. 4e & 4d: What is the difference for two figures? time? How much?
Time axes need scaling...pulse frames for 1's can be separated by dotted lines...the output does not seem sync with the inputs in Fig. 5.
Why is loss not considered in FDTD (Line 167)?
Is there any optimum length for the device? why about 30h? random? 
Why are the pulse width and interval 2 ps and 15 ps, respectively? random? Is there any limit? what about the bit rate? Explain.
Does the system work with chirped or asymmetric pulses?
Give a typical example with realistic dimensions and express the required pulse energy or pump powers for converting 1 to 0. Is it comparable with counterparts?

Author Response

The literature review in Introduction can be improved. What about other linear and nonlinear techniques?
advantages and disadvantages?

We start the Introduction with the review of the literature. Namely, we mention about the solitonic behavior in many branches of physics, electronic and biology [1-3]. We also mention about several theoretical and numerical techniques developed so far to analyze the formation and the propagation of solitons (kinksolitons, spatial and spatio-temporal solitons, gap solitons, etc.) in different kind of media [4]. As in the linear regime, we are giving the details about our original method developed in [44] and [46]. The advantage of this method is its high computation speed and accuracy.

It is said: (In order to realize the propagation of solitons, the one- or quasi-one dimensionality of the medium
is required). Elaborate on this.

The main requirement for the propagation of solitons is the one- or quasi-one dimensionality of the medium
except for the spatial soliton for which a beam propagation direction plays the role of time. In photonics a quasi-one dimensionality can be achieved either by sharp refractive index change between the guiding layer and the surrounding medium, or by the photonic crystals along (PhC fiber) or perpendicular (PhC waveguide) to the direction of the soliton propagation.

What about other nonlinearities in silicon? Nonlinear loss, two photon absorption?

We have not taken into account two photon absorption in our analysis.

It is said: (Note that the second-order susceptibility term χ(2) = 0 [50]). Why? Is it really zero or small? Note
you have a structured silicon.

In our analysis we assume that the second-order susceptibility χ(2) = 0, which applies for centrosymmetric crystals such as silicon. The details are given in [50].

Is the equation correct in the inset of Fig.1 (Right)?

We have checked the equation and we believe that it is correct.

In Eq. 15 what about the loss, higher-order dispersion terms and other 3rd-order nonlinearities (Raman and
so on)?

We have not taken into account the higher-order dispersion terms in our analysis.

Fig. 3b: why is Lambda different for two cases? Is it a very good agreement (line 142)?

Lambda reflect the results based on the theory and numerical experiment. Please note that in the numerical experiment the leakage in the transverse directions, the interaction between the rods, the back-reflection, and etc. are accounted, whereas the theoretical model describes an “ideal” case. So that, taking into account a complexity of the structure, the authors believe that the theoretical and numerical results can be considered to be in a very good agreement.

Fig. 4e & 4d: What is the difference for two figures? time? How much?

You are correct. The figures show the magnetic field distribution at different time steps.

Time axes need scaling...pulse frames for 1's can be separated by dotted lines...the output does not seem
sync with the inputs in Fig. 5. Why is loss not considered in FDTD (Line 167)?

You are correct. In the numerical investigations we do not take into account the PhC propagation loss.
However, based on experimental data found in [18], we can conclude that the propagation loss inside a
single PCW is only about 8-9%. Please see page 9.

Is there any optimum length for the device? why about 30h? random?

Our goal was to realize not only functional, but also compact all-optical logic gates. Our numerical
investigations have shown that the optimum length of device is 30*h, where h is a period of the structure
(usually the experimental groups take h around 300-400 nm).

Why are the pulse width and interval 2 ps and 15 ps, respectively? random? Is there any limit? what about
the bit rate? Explain.

The parameters are not randomly taken. There are no limits as well. These parameters come from extensive
numerical analysis and from several numerical tests to properly realize all-optical logic gates.

Does the system work with chirped or asymmetric pulses?

Yes, these pulses can be used in the analysis. The main point is to realize the effect of “perfect digitalization”. The details are given in Introduction.

Give a typical example with realistic dimensions and express the required pulse energy or pump powers for
converting 1 to 0. Is it comparable with counterparts?

In our studies, we have considered the experimentally feasible material, such as air-hole type photonic
crystals. All the parameters are taken from the works of experimental groups. Also please note that we are
working with the non-dimensional parameters and it gives us additional freedom and flexibility to choose the parameters for experiment. As for the parameters of the injected signal, they are written on page 7.

Reviewer 4 Report

In this work, the authors summarizes their previous findings regarding propagation characteristics of band-gap temporal solitons in photonic crystal waveguides with Kerr-type nonlinearity and a realization of functional all-optical NOT, AND, NAND logic gates. As a review paper, it is suggested to add some relevant research results of other researchers.

Author Response

We thank the Reviewer for the comment. We have proposed complete theoretical and numerical analyses
for the realization of functional all-optical logic-gates. We believe that we have demonstrated the novelty of
our analysis comparing to the studies by other groups and also properly cited the fundamental works of other groups. The final step of this work is experiment and the authors hope that this paper might motivate the experimental groups and they succeed in the realization of the all-optical logic gates based on the proposed "perfect digitalization" effect. We have added a few sentences in the Conclusion.

This manuscript is a resubmission of an earlier submission. The following is a list of the peer review reports and author responses from that submission.